

# Assessment of island beach erosion due to sea level rise: The case of the Aegean Archipelago (Eastern Mediterranean)

Isavela N. Monioudi[1], Adonis F. Velegrakis[1], Antonis E. Chatzipavlis[1], Anastasios Rigos[1,2], Theophanis Karambas[3], Michalis I. Vousdoukas[4,1], Thomas Hasiotis[1], Nikoletta Koukourouvli[5], Pascal Peduzzi[6], Eva Manoutsoglou[1], Serafim E. Poulos[7], and Michael B. Collins[8]

[1]Department of Marine Sciences, University of the Aegean, University Hill, Mytilene, GR-81100, Greece
[2]Department of Cultural Technology and Communication, University of the Aegean, University Hill, Mytilene GR-81100, Greece
[3]Department of Civil Engineering, Aristotle University of Thessaloniki, University Campus, GR-54124 Thessaloniki, Greece
[4]European Commission, Joint Research Centre (JRC), Directorate for Space, Security & Migration, Disaster Risk Management Unit, Via E. Fermi 2749, Ispra (VA), I-21027, Italy
[5]Department of Geography, University of the Aegean, University Hill, GR-81100 Mytilene, Greece
[6]UNEP/DEWA/GRID-Geneva, International Environment House, 11 Chemin des Anemones, CH-1219 Chatelaine, Switzerland
[7]Faculty of Geology and Geoenvironment, National and Kapodistrian University of Athens, Panepistimioupoli Zografou, 15784 Athens, Greece
[8]Plentziako Itsas Estazioa, University of the Basque Country, Areatza z/g. E-48620, Plentzia- Bizkaia, Spain

*Correspondence to*: Michalis I. Vousdoukas (michalis.vousdoukas@jrc.ec.europa.eu)

**Abstract.** The present contribution constitutes the first comprehensive attempt to (a) record the spatial characteristics of the beaches of the Aegean Archipelago (Greece), a critical resource for both the local and national economy; and (b) provide a rapid assessment of the impacts of the long- term and episodic sea level rise (SLR), under different scenarios. Spatial information and other attributes (e.g. presence of coastal protection works and backshore development) of the beaches of the 58 largest islands of the Archipelago were obtained on the basis of remote-sensed images available in the web. Ranges of SLR-induced beach retreats under different morphological, sedimentological and hydrodynamic forcing and SLR scenarios were estimated, using suitable ensembles of cross-shore (1-D) morphodynamic models. These ranges, combined with empirically-derived estimations of wave run up-induced flooding, were then compared with the recorded maximum beach widths, to provide ranges of retreat/erosion and flooding at the Archipelago scale. The spatial information shows that the Aegean beaches may be particularly vulnerable to mean (MSLR) and episodic SLRs due to: (i) their narrow widths (about 59 % of the beaches have maximum widths < 20 m); (ii) their limited terrestrial sediment supply; (iii) the substantial coastal development and (iv) the limited existing coastal protection. Modeling results indeed project severe impacts under mean and episodic SLRs, which by 2100 could be devastating. For example, under MSLR of 0.5 m (RCP4.5), a storm-induced sea level rise of 0.6 m is projected to result in complete erosion of between 31 and 88 % of all beaches (29 - 87 % of beaches currently fronting coastal infrastructure and assets), at least temporarily. It appears that, in addition to the significant effort and financial resources required to protect/maintain the critical economic resource of the Aegean Archipelago, appropriate coastal 'set-back zone' policies should be adopted and implemented.



# 1 Introduction

Beaches are critical components of the coastal zone; not only are they significant habitats in their own right (e.g. Defeo and McLachlan, 2013), but also provide protection from marine flooding to other transitional ecosystems and the coastal assets, infrastructure and activities they front (e.g. Neumann et al., 2015). At the same time, tourism has been increasingly

associated with beach recreational activities according to the dominant 'Sun, Sea and Sand-3S' tourism model (Phillips and Jones, 2006). Consequently, beaches have become very important economic resources (Ghermandi and Nunes, 2013), forming one of the pillars of tourism, an economic sector that contributes an estimated 5 % of Global Gross Product - GGP, and about 6 – 7 % of global employment (directly and indirectly) (Hall et al., 2013).

Beaches are also very dynamic environments, controlled by complex forcing-response processes that operate at various

spatio-temporal scales (Short and Jackson, 2013). They are generally under erosion (Eurosion, 2004; IPCC SREX, 2012; IPCC, 2013), which can be differentiated into (a) long-term erosion, i.e. irreversible retreat of the shoreline, due to mean sea level rise (MSLR) and/or negative coastal sedimentary budgets that force either beach landward migration or drowning (Nicholls and Cazenave, 2010); and (b) short-term erosion, caused by storm surges and waves, which may, or may not, result in permanent shoreline retreats but can be nevertheless devastating (e.g. Smith and Katz, 2012; UNECE, 2013). The

accelerating MSLR coupled with episodic storm events will aggravate the already significant beach erosion with severe impacts on coastal activities, infrastructure and assets (e.g. Jiménez et al., 2012) and the beach carrying capacity for recreation/tourism (Valdemoro and Jiménez, 2006; McArthur, 2015).

Beach erosion appears to be particularly alarming in islands. Island beaches are increasingly vulnerable to erosion due to their (generally) limited dimensions and diminishing sediment supply (e.g. Velegrakis et al., 2008). At the same time, island

beaches are amongst the most significant 3S tourism destinations. For example, 3S tourism accounts for more than 23 % of the Gross Domestic Product - GDP in many Caribbean Small Island States - SIDS and, in some cases, e.g. Antigua and Barbuda, for more than 75 % (ECLAC, 2011). Mediterranean islands are also major tourism destinations; in Greece, most of the hotel capacity and foreign tourist arrivals and earnings are associated with the Greek islands (SETE, 2016).

Under a variable and changing climate, projections on the future evolution of beach morphology are not easy, due to

uncertainties regarding both forcing and beach response (e.g. Short and Jackson, 2013). Nevertheless, beach erosion is amongst the first issues to consider when planning for the sustainable development of the coastal zone, particularly in areas where beaches function as natural 'armor' to valuable coastal infrastructure and assets and/or as significant environments of leisure (e.g. Paula et al., 2013). Therefore, assessments of the beach morphological evolution at different spatio-temporal scales are required, based on advanced numerical, analytical, and/or empirical models constructed and applied by

experienced operators, set up/validated using appropriate field data and backed by expert analysis (e.g. Roelvink et al., 2009; Bosom and Jiménez, 2010; Ding et al., 2013). However, such efforts are usually hampered by the (a) scarcity of relevant information in many coastal areas, and (b) dearth in the necessary human and financial resources (e.g. Parker et al., 2013); this is particularly true when assessments of beach erosion are carried out over large spatial scales. All the same, it is



necessary to assess future beach retreat/erosion and flood risk at large spatial scales, in order to identify 'hot spots' and plan for effective adaptation policies and efficient allocation of resources.

Against this background, the objective of the present study is to assess the erosion and temporary inundation/flood risks of the beaches of the islands of the Aegean Archipelago (Greece) under different scenarios of SLR. Towards this objective,

spatial characteristics such as the area, length, maximum width, orientation, sediments and the presence of coastal works and backshore development of the Aegean beaches were recorded. This information was then used in conjunction with projections from ensembles of cross-shore morphodynamic models to obtain estimates of the ranges of potential beach retreat/erosion and flooding under different MSLRs and storm events.

## 2 Aegean Archipelago beaches: Significance, environmental setting and sea level rise

### 2.1 Significance of the Aegean Archipelago beaches

Aegean Archipelago (Fig. 1) consists of several thousand islands and rock islets, with a combined area of 17550 km2 and total coastline length of about 5880 km (Eurosion, 2004). Few of these islands and islets are populated; less than 70 islands have more than 100 and 45 more than 1000 permanent inhabitants (http://www.statistics.gr/portal/page/portal/ESYE/PAGE-themes?p_param=A2001). Yet, Aegean islands form very significant tourist destinations. 50 % of all Greek hotel beds (and

> 60 % of all 5 star hotel beds) are located in the Aegean Archipelago with 43 % of the foreign arrivals to Greece in 2015 (7.4 out of a total of 17.1 million) arriving at its 11 international airports (SETE, 2016).

In recent years, tourism has become a most significant economic activity in Greece. In 2013, foreign earnings of the Greek tourist industry were about US\$ 16.1 billion (http://www.bankofgreece.gr/Pages/el/Statistics/externalsector/balance/travelling.aspx). As recent studies suggest that for

each 1 € generated by tourism in Greece, an additional 1.2 - 1.65 € is created by related economic activity (a multiplier of, at least, 2.2 see IOBE (2012)), it follows that direct and indirect earnings from tourism may account for up to about 20 % of the country's GDP (and 30 % of the private sector employment). Tourism is even more important for the island (local) economies. For example, in 2012 tourism accounted for about 48 % of the GDP of Crete and 60 % of the GDP of the Cyclades and Dodecanese island complexes (SETE, 2016).

In Aegean Archipelago, 3S tourism is the dominant model. A most critical component of 3S tourism is the availability of beaches that are aesthetically and environmentally sound and retain adequate carrying capacity (e.g. McArthur, 2015; Cisneros et al., 2016). Therefore, management of beach erosion that constitutes a major threat for the Aegean island beaches should be prioritized; a decade-old approximation had suggested that about 25 % of the total coastline of the Aegean islands was already under erosion (Eurosion, 2004).



## 2.2 Environmental Setting

Aegean Archipelago is located at the Aegean Sea, a peripheral sea of the Eastern Mediterranean that covers an area of some 160 x 103 km2, drains high relief basins with a total area of 200 × 103 km2 and is connected to Black Sea through the Dardanelles Straits and to Eastern Mediterranean through the Cretan Arc Straits. Aegean Sea is characterized by irregular
morphology due to complex regional tectonics and comprises different geomorphological units (Poulos, 2009), including: an extensive shelf (N. Aegean Shelf), a tectonic trough (N. Aegean Trough), a central platform (Cyclades Plateau) with large concentration of islands as well as deep basins (some > 2500 m deep) mainly in the South Aegean (Fig.1). Aegean Sea shows complex hydrographic patterns and circulation (e.g. Theocharis et al., 1993) which are partly controlled by the cold and low salinity water inputs from the Black Sea through the Dardanelles Strait and the warm and saline water inputs from
the Levantine Sea through the Eastern Cretan Arc Straits (Skliris et al., 2011). Under certain conditions (the Eastern Mediterranean Transient -EMT), the deep Aegean basins have been observed to be the locations of deep water formation in the Eastern Mediterranean (e.g. Zervakis et al., 2000; Androulidakis et al., 2012).

The complex physiography of the Aegean Archipelago controls its wind and wave climate, which is relatively mild due to the short fetches and durations. Northerly winds (44% frequency of occurrence, Androulidakis et al. (2015)) and waves
(Soukissian et al., 2007; 2008) appear to prevail and, although waves are generally more energetic in winter, there are also energetic events in summer forced by N-NE winds ('the Etesians'). Highly energetic wave events of relatively short duration may also occur, particularly along island straits. Soukissian et al. (2008) suggested as the most energetic areas of the Aegean Archipelago (i) the area to the N-NE of the Cyclades platform (particularly the Mykonos-Ikaria Strait) and (ii) the western and eastern Cretan Arc Straits (Fig.1); for example, maximum wave heights of about 11 m (Tp of 13.3 s and direction of
345o N) have been reported for the Mykonos-Ikaria Strait in 22/01/2004. Analysis of ERA-INTERIM wave information (1979-2013) from different representative areas of the Aegean Archipelago carried out as part of the present study shows: (a) mean significant wave heights (Hs) of about 1 m in all areas, apart from an area to the northeast of eastern Cretan Arc Straits (mean Hs of about 0.8 m); (b) mean maximum wave heights of about 2.4 m; and (c) significant interannual variability.

Recent studies on the future wave climate of the Aegean Archipelago, project small changes in significant wave heights for
the 21st century. For the period 2001-2049, significant wave heights in the N. Aegean are projected to slightly increase for the SW waves relative to the 1950 - 2000 reference period, whereas for the end of the century (2050 – 2099), wave occurrence and height patterns are projected to show high spatio-temporal variability (e.g. Prinos, 2014; Tsoukala et al., 2016).




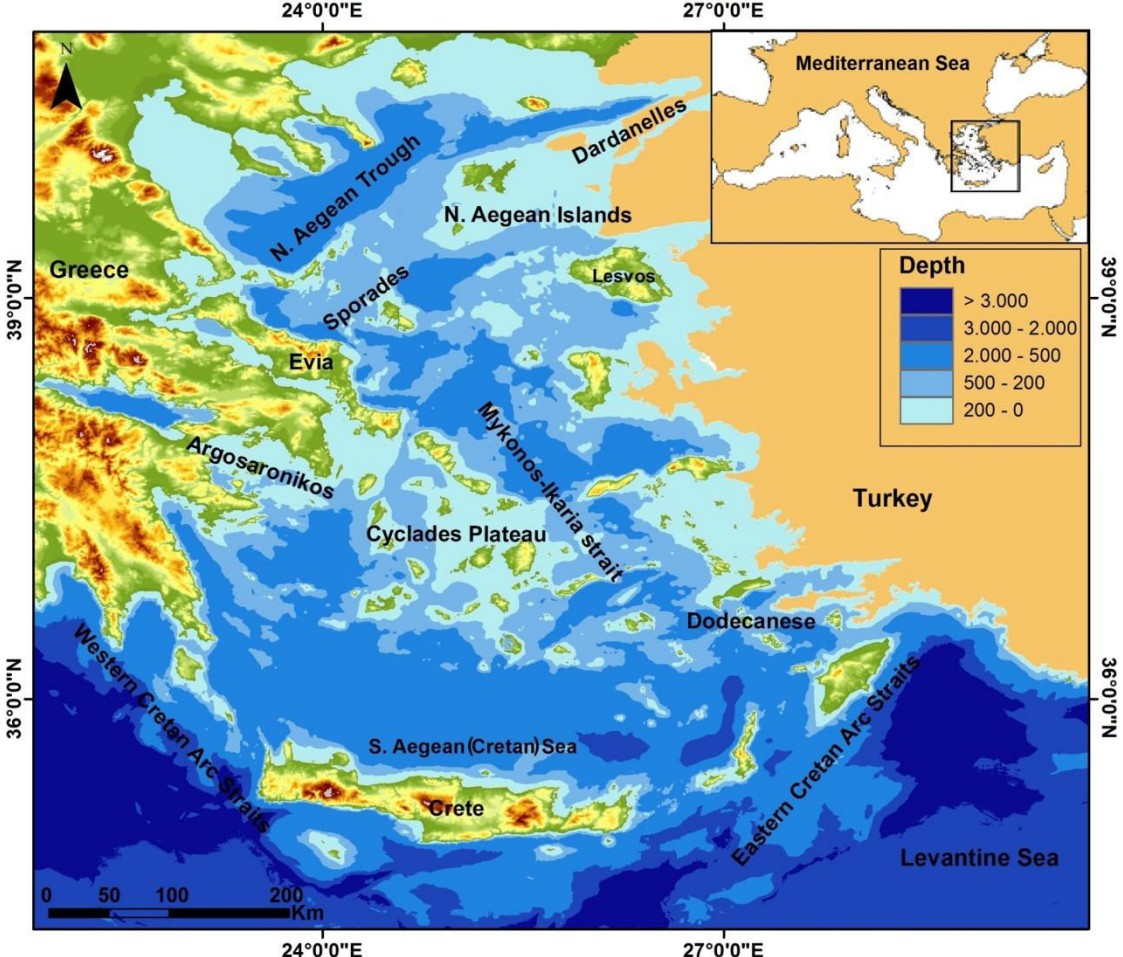

**Figure 1: The Aegean Archipelago**

### 2.3 Mean and  Extreme Sea Levels

Mediterranean MSLR rates were 1.1 – 1.3 mm yr-1 for the most part of the 20th century; since the late 1990s, however,

5   much higher rates (2.4 - 3.8 mm yr-1) have been recorded, an increase attributed mainly to additional water mass inputs rather than to steric contributions being also correlated with North Atlantic Oscillation-NAO modulations (Tsimplis et al., 2013). For the Aegean Archipelago, satellite altimetry suggests recent MSLR rates of 4.3 - 4.6 mm yr-1 (Mamoutos et al., 2014), with some periods characterized by even higher rates (up to 15.3 mm yr-1, see Tsimplis et al. (2009)). In terms of future projections, recent studies project decreases in the MSLR rates in the Aegean Sea for the 21st century; nevertheless,

10   such trends could be underestimations due to the uncertainties regarding water mass exchanges particularly between the Black and Aegean Seas (Mamoutos et al., 2014). Hinkel et al. (2014) using an approach that accounts for changes in ice mass and ocean circulation suggested the following likely future MSLRs for the region of the Aegean Archipelago (33.50-400 N, 18.50-28.50 E). In 2050, mean sea level is projected to be 0.13 - 0.15 m and 0.14 - 16 m higher than that of the 1985



– 2005 reference period for a medium land-ice scenario and RCPs 4.5 and 8.5, respectively; in 2100, under the same scenarios, mean sea level is projected to be 0.46 - 0.48 m and 0.66 - 0.72 m higher than that of the 1985–2005 reference period, respectively.

In addition to MSLR, changes in the intensity, frequency and/or patterns of extreme storm surges and waves can, at least

temporarily, induce beach erosion and flooding, particularly when combined with increasing mean sea levels (e.g. Xu and Huang, 2013). Extreme sea levels in Mediterranean have a seasonal footprint with extreme positive levels occurring mostly in winter and under certain North Atlantic Oscillation (NAO) modulations (Tsimplis and Shaw, 2010). In the Aegean Archipelago, extreme sea levels are relatively low (heights of up to about 0.5 m) (Tsimplis and Shaw, 2010; Krestenitis et al., 2011), increasing slightly towards the north (Androulidakis et al., 2015).

Future extreme sea levels will be associated with high spatial variability, being sensitive to the evolution of the thermohaline circulation and the Black Sea buoyant inputs (e.g. Mamoutos et al., 2014). Extreme levels are projected to show generally decreasing trends over the Mediterranean basin towards the end of the 21st century (e.g. Conte and Lionello, 2013); nevertheless, model choice/resolution (Marcos et al., 2011; Androulidakis et al., 2015; Vousdoukas et al., 2016) and the quality/resolution of the available coastal observations for model validation (Calafat et al., 2014) may have influenced such

projections. Storm surges are generally projected to show (generally) small height increases until 2050, as well as changes in their temporal distribution (e.g. Androulidakis et al., 2015; Vousdoukas et al., 2016).

## 3 Materials and Methods

### 3.1 Geo-spatial characteristics of the Aegean Archipelago beaches

The geo-spatial characteristics of Aegean Archipelago ('dry') beaches have been recorded, on the basis of the images and

other related optical information available in the Google Earth Pro application. In this study, 'dry' beaches were defined as the low-lying coastal sedimentary bodies bounded on their landward side by either natural boundaries (vegetated dunes and/or cliffs) or permanent artificial structures (e.g. coastal embankments, seawalls, roads, and buildings) and on their seaward side by the shoreline, i.e. the median line of the foaming swash zone shown on the imagery. Regarding the lateral extent of individual beaches, these were delimited by natural barriers, such as rock promontories. Tiny beaches (areas less

than about 20 m2) were ignored/not included in the data set. Digitization of the remote-sensed imagery was carried out by few (4) analysts, who followed consistently the above beach delimitation rules. To assess inconsistencies, 400 beaches from different islands (about 12 % of the recorded beaches) were processed by all 4 analysts and the standard deviation of the extracted shoreline positions was estimated less than 0.3 m; this was considered acceptable for the scope of the study.

Beaches were digitized as polygons and exported to a GIS for further analysis. There has been no geo-rectification, as the

aim of the exercise has not been to provide definitive locations and elevations of beach features, but to extract/record (horizontal) geo-spatial characteristics. To this end, a custom-made AML (ARC Macro Language, proprietary language for ArcInfo applications in ESRI software) script was used to estimate beach areas, lengths, maximum widths and orientations



(Allenbach et al., 2015). It should be noted that the satellite imagery used cannot provide synoptic information at the Aegean Archipelago scale, as images have been collected in different years, seasons and hydrodynamic conditions. Although tidal effects are small due to the microtidal regime of the Aegean Archipelago (tidal ranges in most areas less than about 0.15 m) and the (generally) increased beach slopes, geo-spatial characteristics controlled by the shoreline position and obtained from remotely-sensed snapshots may not represent mean conditions (Velegrakis et al., 2016). Nevertheless, this cannot be avoided when working at a basin/Archipelago scale (Allenbach et al., 2015).

In addition to beach dimensions, other relevant information was recorded and codified, including: the presence of (a) natural features, such as river mouths, back-barrier lagoons and cliffs and beachrock outcrops; and (b) artificial features such as coastal protection schemes and backshore infrastructure/assets. Assessment of the beach sediment texture (e.g. sand or gravel) was also carried out on the basis of web-based optical information and other available information collated from scientific literature/reports.

## 3.2 Beach retreat predictions due to sea level rise

Sea level rise represents a most significant threat to beaches, forcing their retreat/erosion; a sea level rise α will result in a shoreline retreat S due to erosion of the beach face, the sediments of which are transported/deposited offshore, with the extent/rates of the cross-shore retreat controlled (amongst others) by bed slope, the texture and supply of beach sediments and the hydrodynamic conditions (e.g. Dean, 2002).

In the present study, seven cross-shore morphodynamic models were used to project beach response to SLR: the Bruun (Bruun, 1988), Edelman (Edelman, 1972) and Dean (Dean, 1991) analytical models and the numerical models SBEACH (Larson and Kraus, 1989), Leont'yev (Leont'yev, 1996), XBEACH (Roelvink et al. 2010) and a model, the hydrodynamic component of which involves high-order Boussinesq equations- Boussinesq model (Karambas and Koutitas, 2002). The Bruun model is a widely-used (e.g. Hinkel et al., 2010; Ranasinghe et al., 2013) analytical morphodynamic model that estimates long-term coastal retreat S under a SLR a on the basis of the equilibrium profile concept; its results are controlled by the height of the beach face and the cross-shore distance between the beach closure depth and the shoreline. The Edelman model estimates beach erosion/retreat using the initial height of the beach face, the water depth at wave breaking and the surf zone width, whereas the Dean model estimates retreats on the basis of the wave height, the water depth at wave breaking and the surf zone width.

The SBEACH model (Larson and Kraus, 1989) is a numerical morphodynamic model, consisting of 3 modules: a hydrodynamic, a sediment transport and a morphological evolution module. It can describe wave transformation in shoaling waters, with the coastal sediment transport controlled by the coastal wave energy fluxes; the sediment continuity equation in a finite difference scheme and a 'stair–step' beach profile discretization is used in its morphological module. The numerical model based on Leont'yev (1996) uses the energetic approach, with the cross-shore wave energy balance controlled by wave propagation angle and dissipation; sediment transport rates are estimated separately for the surf and swash zones. The XBEACH model (Roelvink et al., 2010) is an open-source, widely used numerical model of the nearshore processes intended





to estimate the effects of time-varying storm conditions; it contains a time-dependent wave action balance solver and allows for variations in the wave action over time and over the directional space. Finally, the Boussinesq model used computes non-linear wave transformation in the surf and swash zone, based on a wave propagation module involving high-order Boussinesq equations (Karambas and Koutitas, 2002); its sediment transport module can estimate sheet flow as well as bed

and suspended load over uneven sea beds (e.g. Karambas, 2006). Detailed descriptions of the models used can be found elsewhere (e.g. Vousdoukas et al., 2009a; Monioudi et al., 2014).

In the present contribution, all models were used in a stationary mode. Validation of model results was provided through comparisons with the results of physical experiments in the GWK wave flume, Hanover, Germany (see Section 4.2.1). The models were used in an ensemble mode in order to assess the range of long- and short-term beach retreats/erosion for

different beach slopes, sediment textures (grain size) and wave conditions, and under different scenarios of MSL changes and/or extreme sea levels caused by storm surges/waves. Two model ensembles were created, a 'long-term' ensemble consisting of the analytical models Bruun, Dean and Edelman and a 'short-term' ensemble comprising the numerical SBEACH, Leont'yev, XBEACH and Boussinesq models; the former is used to assess beach retreat/erosion under MSLR, whereas the latter retreat due to temporary SLR i.e. from storm surges/waves. The adopted approach was based on the

proposition that as models have differential sensitivity to the controlling environmental factors, ensemble applications may provide 'tighter' prediction ranges than the individual models (Section 4.2.1).

Experiments were carried out using various plausible wave conditions in the Aegean Archipelago (Section 2.2), i.e. waves with offshore heights (Hs) of 1, 1.5, 2, 3 and 4 m and periods (T) of 4, 5, 6, 7 and 8 s. Likewise, in order to address the sediment texture variability over the Archipelago beaches, experiments were carried out for seven different median (d50)

grain sizes (d50 of 0.2, 0.33, 0.50, 0.80, 1, 2 and 5 mm); note that the results of the analytical models are not controlled by beach sediment size. Five (5) different linear profile slopes (bed slopes of 1/10, 1/15, 1/20, 1/25 and 1/30) and twelve (12) SLR scenarios (0.05, 0.15, 0.22, 0.30, 0.40, 0.50, 0.75, 1, 1.25, 1.50 and 2 m) were examined. Experiments were carried out for all combinations (about 5500 experiments), and the means (best fits) of the lowest and highest projections by all models of the two ensembles were estimated. With regard to combined SLRs (i.e. storm surges superimposed on MSLRs), the long-

term and short-term ensembles were used consecutively.

Recent MSLR projections for the area that take into account the contribution of ice mass were used (Hinkel et al., 2014): (i) 0.15 m, average of the RCP 4.5 and RCP 8.5 scenarios in 2040; and (ii) 0.5 m and 0.7 m under RCP 4.5 and RCP 8.5 (2100), respectively. With regard to short-term SLR, recent trends/projections for storm surge heights in the Aegean Archipelago (up to 0.5 - 0.6 m (Tsimplis and Shaw, 2010; Androulidakis et al., 2015; Vousdoukas et al., 2016), were used.

The above approach is designed to project beach retreat/erosion, but not temporary inundation/flooding due to wave run-up. Although the wave run–up is dealt within the numerical models of the ensemble, its effects are manifested in the results only if it induces sediment transport that forces morphological changes (e.g. Leont'yev (1996)). Yet, wave run up-induced temporary flooding that does not result in beach retreats is also a significant management issue (e.g. Jiménez et al., 2012; Hoeke et al., 2013). Therefore, estimations of wave run up excursion/inundation were also undertaken on the basis of run up



heights; these were estimated for all tested conditions, using the expressions of Stockdon et al. (2006) which have been validated for the beaches of the Aegean Archipelago (Vousdoukas et al. 2009b):

$$R_{2\%} = 1.1 \left( 0.35\beta(H_o/L_o)^{1/2} + \frac{[H_oL_o(0.563\beta^2+0.004)]^{1/2}}{2} \right), \quad \text{(all data)} \tag{1}$$

$$R_{2\%} = 0.043(H_o/L_o)^{1/2}, \text{ for dissipative beaches } (\xi < 0.3) \tag{2}$$

where $R_{2\%}$, the 2% exceedence of the peak run-up height, *Ho, Lo* are the deep water wave height and length, *β* the beach slope and *ξ* the Iribarren number ($\xi = \beta/(Ho/Lo)^{1/2}$).

Wave run up excursions were then calculated from the wave run up heights ($R_{2\%}$) for all tested bed slopes and wave conditions and added to the beach erosion/retreat projections of the seven 1-D cross-shore morphodynamic models to project final flooding excursions (*S(i)*). The best fits of the lower and upper limits of the final projections of flooding by all models were then estimated.

## 4 Results

### 4.1 Characteristics of the beaches of the Aegean Archipelago

3234 beaches were recorded along the coasts of the 58 larger islands of the Aegean Archipelago. These beaches were found to have a total area of about 21.35 km$^2$, indicating that the total carrying capacity of the Aegean Archipelago beaches (i.e. the number of visitors that can be simultaneously hosted), is about 2135 thousand according to the Rajan et al. (2013) criterion (10 m$^2$ for each beach user). A rough estimation on the basis of the average number of tourist days spent in the Aegean islands suggests that, with the exception of the busiest months (July and August) during which about 7000 thousand tourist days per month are recorded (SETE, 2016) and many beaches operate close to their full carrying capacity, the Archipelago beaches in their present condition still have (as a total) potential for development as environments of leisure.

Most of the Aegean beaches are narrow, with about 59 % of them having maximum widths of less than 20 m (Fig. 2). Regarding beach sediment type, three different beach types were recorded: (i) sandy beaches (36.4 %); (ii) gravel/pebble beaches (44.4 %); and (iii) cobble/rocky beaches (about 2 %). For the remainder of the beaches, no sediment type could be assigned on the basis of the available information (Fig. 2).





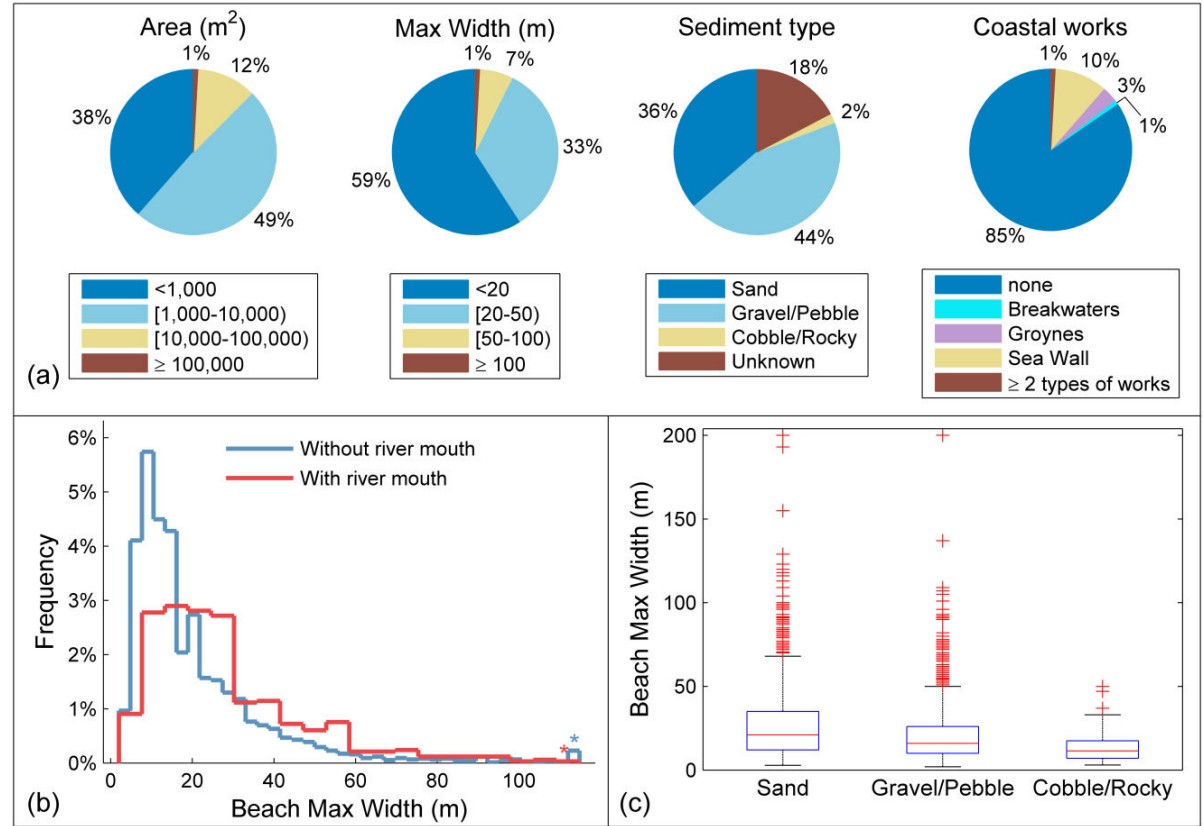

**Figure 2: Statistics of the Aegean Archipelago beaches. (a) Pie diagrams of major characteristics; (b) relationship between the beach maximum width and the presence of river mouth; and (c) boxplots showing relationships between beach widths and sediment types.**

Beach orientation was examined by: (i) taking into consideration the 8 main compass directions; and (ii) considering the 8 main compass directions for each of the 8 island complexes of the Aegean Sea (N. Aegean, Dodecanese, Cyclades, Crete, SW Aegean, Argosaronikos, Evia, and Sporades, Fig. 1). In the first case, occurrence of beaches with S, SE and SW orientations appears more prevalent (16.1 %, 13.9 % and 12.5 % respectively) than that of beaches facing towards the other 5 compass directions; this was found to be statistically significant (Table 1). Prevailing beach orientations were also found to differ over the different island complexes. For example, the prevailing (statistically significant) beach orientations in the Dodecanese islands were found to be towards the SE (18 %) and in Crete towards the S (22%). There is also significant correlation between beach orientation and maximum width. Statistical tests (Table 1) showed that W and SW facing beaches are associated with greater maximum widths (24.5 and 24.0 m, respectively).

**Table 1**: **Statistical analysis of the characteristics of the Aegean Archipelago beaches.**





| Comparison | Null Hypothesis ($H_0$) | Statistical test results | Observations |
|---|---|---|---|
| Beach orientation and occurrence | $H_0$: The data are uniformly distributed around the compass rose | *Rayleigh's Test*: $H_0$ rejected (p<0.001). Beach orientation not uniformly distributed over the compass rose | Prevailing beach orientation towards the south (S) (occurrence 16.1 %) |
| Beach orientation and maximum width | $H_0$: The 8 compass orientations have equal mean values regarding maximum width $(\mu_1 = \mu_2 = ... = \mu_8)$ <br><br> $H_0$: $\rho_M = 0$, No correlation in the data population | *Kruskal-Wallis (Analysis of variance by ranks):* $H_0$ rejected (p < 0.001). Mean beach maximum widths differ with orientation <br> *Mardia's correlation coefficient (Zar, 2010):* $H_0$ rejected (p < 0.01). Significant correlation between orientation and maximum width | W and SW facing beaches have greater maximum widths (mean maximum widths of 24.5 and 24 m) |
| Beach orientation and beachrock occurrence | $H_0$: the two variables are independent | $\chi^2$ tests for (i) 4 and (ii) 8 main compass orientations*: $H_0$ rejected (p < 0.001). Small association's power Cramer's (i) V = 0.13 and (ii) V= 0.17 | No significant trend |
| Sediment type and presence of river mouth | $H_0$: Beach sediment type independent of the river mouth presence | $\chi^2$ test: $H_0$ rejected (p < 0.05). Significant correlation, but small power of association (Cramer's V= 0.064 ) | Beaches with coarse grained sediments are less associated with river mouths |
| Beach maximum width and presence of river mouth | $H_0$: the sets are independent (mean values $(\mu_1 = \mu_2)$ | *Student's t-test* (two-tailed test)*: $H_0$ rejected (p < 0.001) | Beaches with large maximum widths are associated with river mouths (Fig. 2b) |
| Beach sediment type and maximum width | $H_0$: Mean values of the 3 categories are equal $(\mu_1 = \mu_2 = \mu_3)$ | *ANOVA,* after splitting the data set into the 3 sediment types: $H_0$ rejected (p <0.001). | Beaches with coarse sediments are associated with smaller maximum widths (Fig. 2c) |

The above results indicate some hydrodynamic control in the development and maintenance of the Aegean beaches: there are more and wider beaches along the island coasts that are (at least) partially protected from the prevailing northerly wind/waves (Soukissian et al., 2007; 2008). However, the correlation is weak (Table 1) due to other important factors

5    controlling beach development and maintenance (e.g. the antecedent topography and geological history, the terrestrial sediment supply, human development and the wave fetch/duration at each individual beach).

In terms of terrestrial sediment supply, few beaches (about 18 %) were found to be associated with intermittent (very rarely permanent) flow river mouths; most of those were found in the large islands of North Aegean (314 out of a total of 728 beaches). Riverine supply appears to be a significant control for both beach width and sediment type; beaches with river

10    mouths are more likely to be wider and built on finer sediments (sands) (Fig. 2 and Table 1).

Another interesting finding is that perched beaches (Gallop et al., 2012) form a significant fraction of the Aegean Archipelago beaches; beachrock outcrops (e.g. Vousdoukas et al., 2009a) were recorded on the beach face of 23 % of all beaches. In terms of orientation, although there appears that beaches with beachrocks are more prevalent at the southern coasts of islands (192 out of 744 total occurrences), statistical testing did not show any significant trends (Table 1). As there





could be a significant number of beaches which may contain buried beachrocks (i.e. not outcropping at the time of the analysed imagery), it seems that beachrocks are quite widespread along the Aegean island beaches.

With regard to coastal protection schemes, these are present only at about 15 % of the Aegean island beaches (Fig. 2a). In comparison, many beaches are associated with coastal development: 80.8 % of all beaches front public and private assets
such as coastal roads, housing and tourist infrastructure. The density of these assets is variable, ranging from a coastal road and few houses found in remote island beaches to the plethora of valuable public and private assets found behind the urban beaches of e.g. Heraklion (Crete) and Rhodes. Generally, about 32.7 % of the Aegean beaches front coastal infrastructure and assets with moderate/high density. The considerable coastal development, coupled with the narrow widths of the Aegean island beaches increase exposure under a variable and changing climate.

**4.2 Predictions of erosion/retreat and flooding for the Aegean island beaches**

**4.2.1 Model sensitivity and validation**

Model sensitivity tests undertaken within the present study have shown that beach retreats are controlled by beach typology. All models show higher retreats with decreasing Iribarren numbers $\xi$ (i.e. for beaches with milder beach slopes $\beta$ and/or offshore waves of increased steepness ($H_o/L_o$), with the exception of Bruun model the results of which are independent of $\xi$
for linear profiles. The most sensitive models to beach slope are the Edelman and SBEACH models and the least sensitive the Xbeach model; with regard to the offshore wave climate, Xbeach appears to be the most and Leont'yev and Boussinesq the least sensitive models. The effect of sediment texture is not always clear, although a weak negative correlation between beach retreat and the median sediment size ($d_{50}$) might be discerned in the numerical models; analytical model results are independent of the beach sediment texture. Generally, models showed differential sensitivity to initial conditions and
forcing, which justifies their collective use in ensembles.

Model results have been compared with those by physical experiments conducted at the wave flume (GWK, Hanover, Germany) in early 2013 (details in Vousdoukas et al., (2014)). In these experiments, the initial slope of the beach was set to about 1/15, tested waves had an offshore height (H) of 1 m and a period (T) of 5 s and the seabed consisted of well-sorted sand ($d_{50}$ = 0.3 mm). Three SLR scenarios were tested (rises of 0.2, 0.4 and 0.6 m), with the initial profiles of these
experiments controlled by the final profile (wave forcing only) of the first experiment (i.e. the test without level rise). Simulation times were set to 3000 s.

In Fig. 3, profiles by the numerical models Leontyev, SBEACH, Xbeach and Boussinesq under the baseline level as well as 3 increased sea levels (+0.2, +0.4, and +0.6 m) are compared against those resulted from the physical experiments. It appears that there are some discrepancies, particularly with regard to the dynamics of the offshore bar and trough. The Leontyev, and
Xbeach models seem to considerably 'smooth' these features, whereas the SBEACH and Boussinesq results tend to represent better the results of the physical experiments; the best performance was by the Boussinesq model (Fig. 3). Nonetheless, there appears to be a good agreement between the beach retreats estimated by the models and those recorded





during the physical experiments (Fig. 3 and Table 2). Leonty'ev, SBEACH and Boussinesq models appear to slightly underestimate and Xbeach to overestimate beach retreats. Bruun and Edelman models appear to overestimate beach retreats under higher SLRs whereas the Dean model under all SLRs tested.

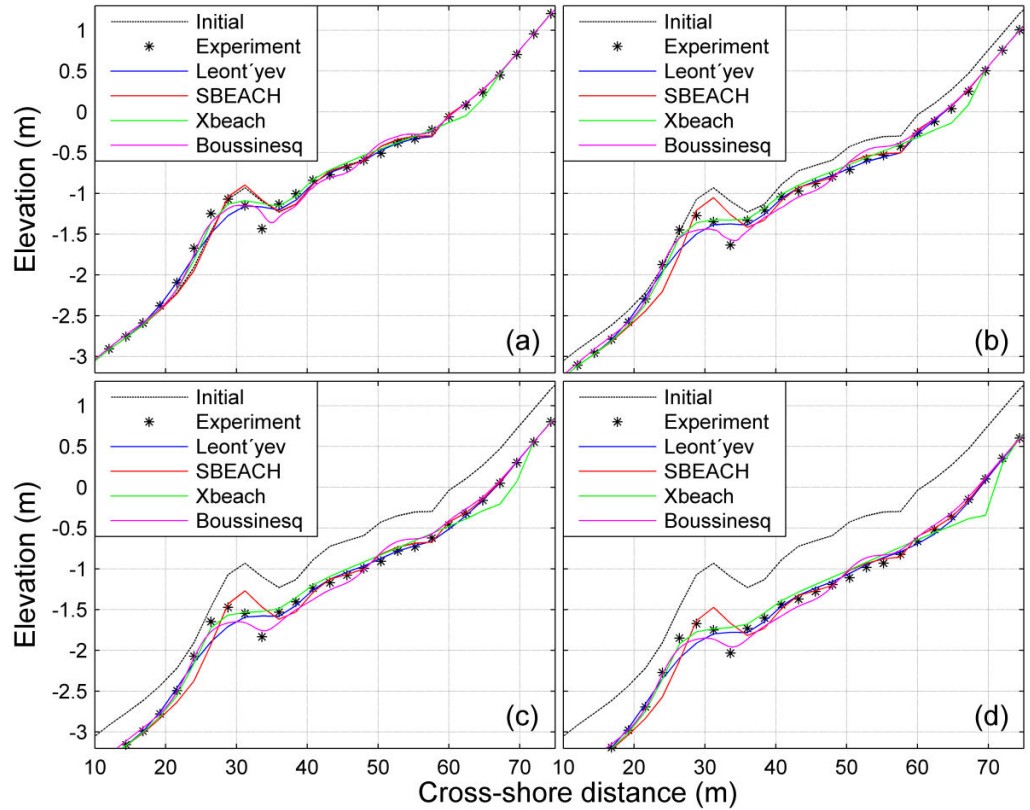

**Figure 3: Profiles by numerical models plotted against results from physical experiments at the GWK wave flume (Hanover): (a) initial/present water level; (b) water level rise of 0.2 m; (c) rise of 0.4 m; and (d) rise of 0.6 m. Both numerical and physical experiments were set up for the same initial (non-linear) profile (stippled line). Simulation times for the numerical experiments set to 3000 s to match those of the physical experiment.**

When the models are used in an ensemble mode, the comparison between the retreats projected by the numerical models (short-term retreat) and those recorded in the physical experiments improves (Table 2). Conversely, analytical models tend to overestimate beach retreats by up to 20 %. Interestingly, when the results of all (7) analytical and numerical models are combined in a unified ensemble, then retreat overestimations decrease to less than 10 % for the tested SLRs.

**Table 2: Comparison of results from models and physical experiments. Negative values represent erosion. Key: SLR, sea level rise; Exper, physical experiment; Leo, Leont'yev model; SB, SBEACH model; Xb, Xbeach model; Bous, Boussinesq model; Br, Bruun model; Edel, Edelman model; Dean, Dean model; ST Ens, Short-term ensemble; and LT Ens, Long-term ensemble.**



| SLR (m) | Shoreline retreat/erosion or advance/accretion (m) | | | | | | | | | |
|---|---|---|---|---|---|---|---|---|---|---|
| | Exper | Leo | SB | Xb | Bous | ST Ens | Br | Edel | Dean | LT Ens |
| 0 | -0.35 | -0.01 | 0.02 | -2.32 | -0.06 | -0.59 | --- | --- | --- | --- |
| 0.2 | -3.57 | -3.2 | -3.04 | -5.63 | -3.0 | -3.72 | -3.01 | -3.09 | -6.7 | -4.27 |
| 0.4 | -6.23 | -5.94 | -5.83 | -8.35 | -5.6 | -6.43 | -6.13 | -6.43 | -8.82 | 7.13 |
| 0.6 | -8.66 | -8.22 | -8.02 | -10.27 | -7.9 | -8.60 | -9.25 | -9.97 | -11.5 | -10.24 |

The above results refer to natural profiles. In order to relate the validation to the linear profiles used here, further tests were undertaken. In these tests, the models were run using an 'equivalent' linear profile having a slope of 1/15 to compare with the results of the physical experiments. The 'equivalent' profile was estimated using the best linear fitting to the natural

profiles used in the physical experiment (between water depths of 3.5 and elevations of 1.5 m).

The results of this exercise are shown in Table 3. Generally, the comparison showed that, at least for the conditions tested, the results of the models set with the equivalent linear profile were reasonably close to those of the physical experiments, with the comparison worsening with increasing water levels. When the numerical models are used as a (short-term) ensemble, beach retreat forced by waves only is substantially overestimated; under SLR, however, ensemble projections

improve (overestimations of up to 19 %, Table 3). Regarding the analytical models, the long-term ensemble gave improved projections, with differences between models and experiments ranging from about 3 % to 11 %.

**Table 3**: **Comparison of results by models and physical experiments. Models used an 'equivalent' linear profile slope of 1/15 to represent the natural profile of the physical experiment (see text). Negative values represent beach retreat/erosion. Key: SLR, sea**
**level rise; Exper, physical experiment; Leo, Leont'yev model; SB, SBEACH model; Xb, Xbeach model; Bous, Boussinesq model; Br, Bruun model; Edel, Edelman model; Dean, Dean model; ST Ens, Short-term ensemble; and LT Ens, Long-term ensemble.**

| SLR (m) | Shoreline retreat/erosion or advance/accretion (m) | | | | | | | | | |
|---|---|---|---|---|---|---|---|---|---|---|
| | Exper | Leo | SB | Xb | Bous | ST Ens | Br | Edel | Dean | LT Ens |
| 0 | -0.35 | -0.34 | -0.02 | -3.23 | 0.0 | -0.9 | --- | --- | --- | --- |
| 0.2 | -3.57 | -3.58 | -3.2 | -7.00 | -3.03 | -4.20 | -3 | -3.07 | -4.05 | -3.37 |
| 0.4 | -6.23 | -6.67 | -6.25 | -10.47 | -5.9 | -7.32 | -6 | -6.3 | -7.05 | -6.45 |
| 0.6 | -8.66 | -9.73 | -9.25 | -13.35 | -8.8 | -10.28 | -9 | -9.69 | -10.05 | -9.58 |

### 4.2.2 Beach erosion and inundation/flooding projections

Modeling results show that sea level rise will cause shoreline retreats as well as significant morphological changes (Fig. 4). Numerical models showed differential profile changes in both breaker and surf zones, supporting their use in an ensemble mode.





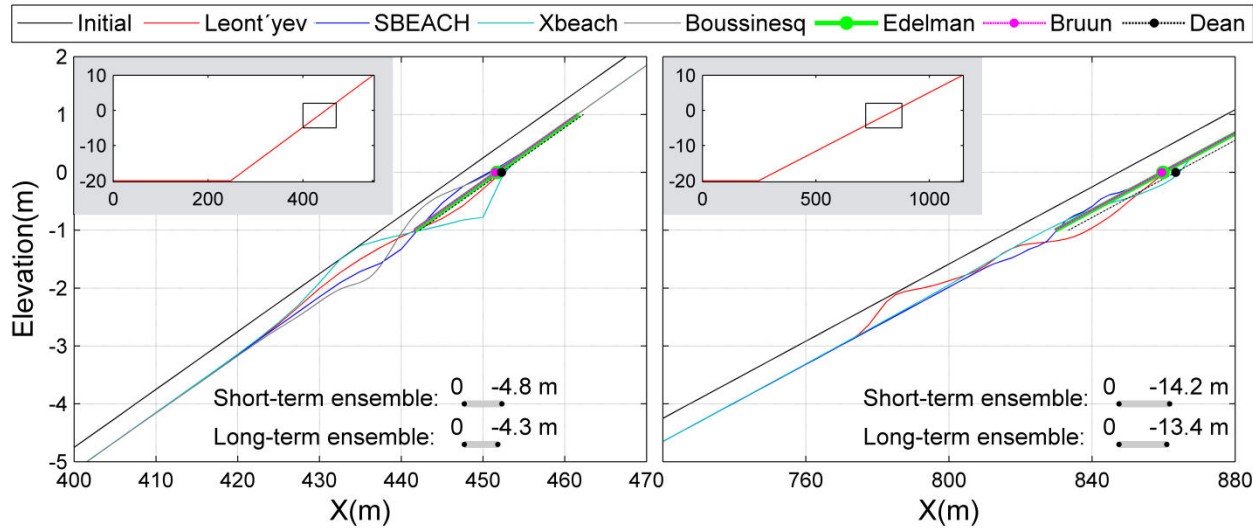

**Figure 4: Examples of morphodynamic changes in the profile of the upper part of the beach (see inset) and mean retreats of the 2 ensembles for SLR of 0.4 m. (a) Offshore (at 20 m water depth) wave height H of 1 m and period T of 5 s, and median (d50) sediment grain size of 0.8 mm; linear beach profile with 1/10 slope; (b) Offshore (at 20 m water depth) wave height H of 2 m and period T of 6 s, and median (d50) sediment grain size of 0.33 mm; linear profile with 1/30 slope. Origin of X axis at 20 m water depth.**

Models displayed differential behavior for almost all tested conditions, showing as expected significant ranges of results (Fig. 5) due to the varying initial conditions and forcing used i.e. different bed slopes, sediment sizes, wave conditions and SLRs. The means (best fits) of the lowest and highest projections of all models were calculated. It was found that the 'low' mean of the beach retreat projections by the short-term ensemble (i.e. the best fit of the lowest projections from the 4 numerical models) is given by $S = 0.1\,\alpha^2 + 9.7\,\alpha + 0.4$ and the 'high' mean by $S = 0.7\,\alpha^2 + 28.5\,\alpha + 4.8$ (Fig. 5a), where S is the beach retreat and $\alpha$ is the SLR. Also, the low projection mean of the long-term ensemble is given by $S = 0.1\alpha^2 + 10\,\alpha + 0.3$ and the high projection mean by $S = 1.6\,\alpha^2 + 29.8\,\alpha + 2.3$ (Fig. 5b). Ranges in beach temporary inundation/flooding due to wave run-up combined with (a) episodic (short term) SLRs (Fig. 5c) were estimated as $S(i) = 0.1\,\alpha^2 + 9.7\,\alpha + 4$ (minimum) and $S(i) = -0.7\,\alpha^2 + 31.2\,\alpha + 27.2$ (maximum) and with long-term SLRs (Fig. 5d) as $S(i) = 0.4\,\alpha^2 + 10.1\,\alpha + 3.7$ (minimum) and $S(i) = 0.4\,\alpha^2 + 30\,\alpha + 28.1$ (maximum).

Ranges of decreases in 'dry' beach widths were projected through the comparison between the ranges of beach retreat/erosion (S) and the maximum widths of the 3234 beaches. Ranges in beach temporary inundation/flooding were estimated by the comparison between the ranges of combined beach retreat and wave run-up excursions (S(i)) and the beach maximum widths. In Table 4, estimations are presented for 9 SLR scenarios: (i) 0.15, 0.5 and 0.7 m MSLRs according to recent projections for the area (Hinkel et al. 2014) using the long-term ensemble; (ii) short-term SLRs due to storm surges/waves of 0.2, 0.4 and 0.6 m using the short-term ensemble; and (iii) combined MSLRs and storm events of 0.55, 1.1 and 1.3 m, using consecutively the long- and short-term ensembles.



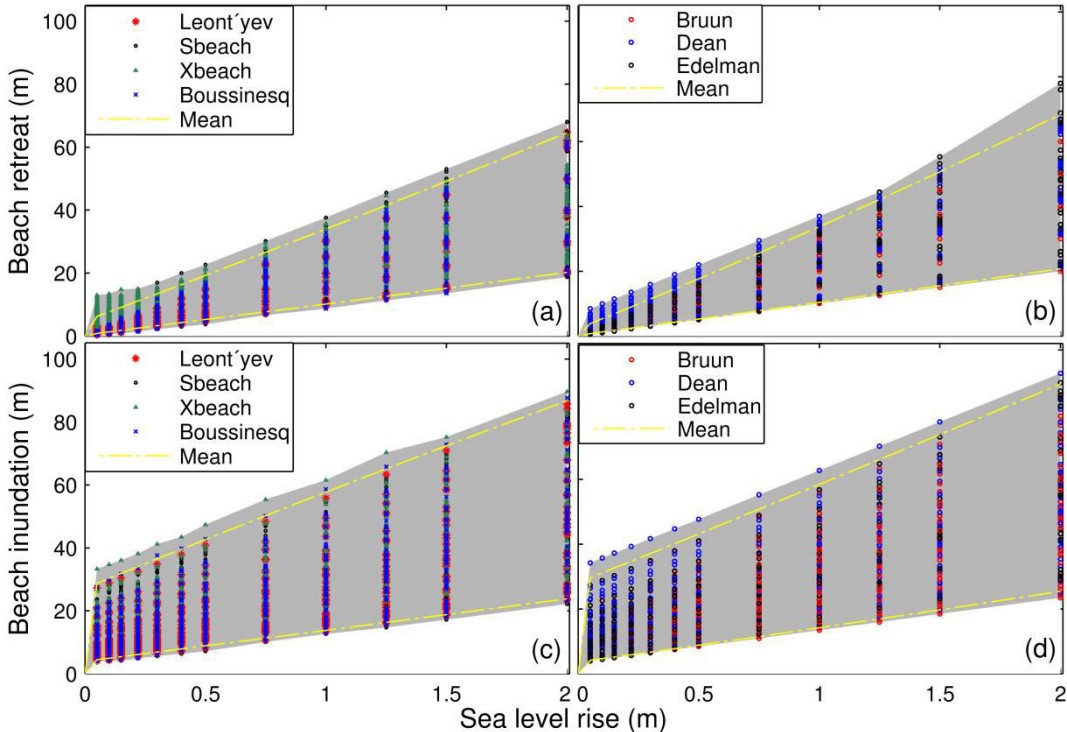

**Figure 5: Projections of beach retreat/erosion (a and b) and temporary inundation/flooding (c and d) due to short-term (a and c) and long-term (b and d) SLR. Projections are for different beach slopes, sediment sizes and wave conditions. The means of the highest and lowest projections of all models in the ensembles are shown as yellow stippled lines.**

According to the projections of the long-term ensemble, MSLRs of 0.15, 0.5 and 0.7 m will result to beach retreats/erosion by about 1.8 - 6.8, 5.3 - 17.6 and 7.3 - 24 m, respectively, whereas short-term ensemble estimates show that episodic increased levels of +0.2, +0.4, +0.6 m could result in ranges of beach retreat/erosion of about 2.3 - 10.6, 4.3 - 16.3, 5.3 - 19.3 and 6.2 - 22.2 m, respectively (Table 4). For the considered SLR scenarios, there will be significant impacts on the Aegean Archipelago beaches as shown by the percentages of beaches that are projected to be eroded/shifted landward to a distance equal to 50 % and 100 % of their maximum width (Table 4).

Even under a MSLR of 0.15 m (RCPs 4.5 and 8.5 in 2040), there could be substantial impacts on the basis of the mean minimum and maximum projections of the long-term ensemble (Table 4). Temporary inundation (S(i)) could overwhelm between 4.9 and 81 % of the beaches, flooding (occasionally) 4.6 - 80 % of the beaches fronting currently existing coastal infrastructure/assets. Under a MSLR of 0.5 m (RCP 4.5, 2100), projected impacts will be severe. 5 to 54 % of all Aegean beaches will be completely eroded in the absence of appropriate coastal defences, whereas between about 18 and 90 % of all beaches are projected to be occasionally overwhelmed by flooding. For 2100, under the high emission scenario (RCP8.5, MSLR of 0.7 m), impacts could be catastrophic (Table 4 and Fig. 6): 12.4 - 68 %  and 28 - 93 % of Aegean island beaches





will be will be completely eroded and occasionally overwhelmed by flooding, respectively. Associated infrastructure/assets are also projected to be greatly impacted, with 12 - 66 % and 26 – 92 % of all beaches fronting currently existing assets projected to be lost to beach erosion and occasionally overwhelmed by flooding, respectively.

**Table 4**: **Minimum and maximum mean estimates of beach retreat (S) and inundation/flooding (S(i)) by the long-term, short-term and combined ensembles and the empirical model of Stockdon et al. (2006) Ranges of cross-shore retreat/erosion (R) and temporary inundation/flooding (F) for the beaches of the Aegean Archipelago are projected by comparing the highest and lowest mean S and S(i) with the maximum width of the 3234 Aegean Archipelago beaches under different SLRs. Numbers (N) and percentages of beaches where backshore infrastructure/assets are projected to be affected by beach retreat/erosion and flooding**
**are also shown.**

| SLR (m) | | | S (m) | S(i) (m) | R | F | R | F | R | | F | |
|---|---|---|---|---|---|---|---|---|---|---|---|---|
| | | | | | Equal to 50 % of max. width (%) | | Equal to max. width (%) | | Beaches with assets affected | | | |
| | | | | | | | | | N | % | N | % |
| Long-term | 0.15 | Min | 1.8 | 5.2 | 0.6 | 27.4 | 0.0 | 4.9 | 0 | 0.0 | 119 | 4.6 |
| | | Max | 6.8 | 32.6 | 39.1 | 96.4 | 8.2 | 81.1 | 203 | 7.8 | 2086 | 79.8 |
| | 0.5 | Min | 5.3 | 8.8 | 27.5 | 54.2 | 5.0 | 17.6 | 120 | 4.6 | 437 | 16.7 |
| | | Max | 17.6 | 43.2 | 84.4 | 98.3 | 54.2 | 89.7 | 1368 | 52.4 | 2320 | 88.8 |
| | 0.7 | Min | 7.3 | 10.9 | 43.2 | 64.3 | 12.4 | 27.6 | 306 | 11.7 | 688 | 26.3 |
| | | Max | 24 | 49.3 | 91.5 | 99.1 | 68.0 | 92.6 | 1732 | 66.3 | 2400 | 91.8 |
| Short-term | 0.2 | Min | 2.3 | 5.9 | 2.4 | 30.9 | 0.1 | 5.0 | 2 | 0.1 | 122 | 4.7 |
| | | Max | 10.6 | 33.4 | 64.3 | 96.6 | 27.5 | 82.1 | 684 | 26.2 | 2113 | 80.9 |
| | 0.4 | Min | 4.3 | 7.9 | 17.4 | 47.9 | 2.3 | 12.7 | 49 | 1.9 | 315 | 12.1 |
| | | Max | 16.3 | 39.6 | 81.1 | 97.8 | 51.3 | 87.2 | 1293 | 49.5 | 2249 | 86.1 |
| | 0.6 | Min | 6.2 | 9.8 | 34.7 | 59.1 | 7.9 | 21.5 | 195 | 7.5 | 537 | 20.6 |
| | | Max | 22.2 | 45.7 | 90.2 | 98.8 | 66.5 | 90.7 | 1695 | 64.9 | 2343 | 89.7 |
| MSLR + Short-term | 0.55 | Min | 6 | 9.6 | 34.7 | 59.0 | 7.9 | 21.4 | 195 | 7.5 | 535 | 20.5 |
| | | Max | 23.2 | 46.4 | 91.0 | 98.8 | 68.0 | 91.0 | 1730 | 66.2 | 2354 | 90.1 |
| | 1.1 | Min | 11.6 | 15.2 | 68.0 | 78.8 | 30.8 | 47.8 | 763 | 29.2 | 1204 | 46.1 |
| | | Max | 40.3 | 63.6 | 97.9 | 99.7 | 88.0 | 96.2 | 2270 | 86.9 | 2506 | 95.9 |
| | 1.3 | Min | 13.5 | 17.2 | 74.6 | 82.8 | 39.1 | 54.1 | 977 | 37.4 | 1364 | 52.2 |
| | | Max | 46.2 | 69.7 | 98.8 | 99.7 | 91.0 | 96.8 | 2354 | 90.1 | 2526 | 96.7 |

According to the mean low projections of the short-term ensemble, increased sea levels due to storm surges/waves of 0.2 and 0.4 m will result in moderate (temporary) beach retreats (0.1 - 2.3 % of all beaches will retreat more than their maximum

width) and temporary flooding (5 – 12.7 % of the beaches will be occasionally completely flooded). On the basis of the mean high projections (forced by the high wave conditions expected in storms, see also Tsoukala et al, (2016)), beach retreats and flooding will be substantial with severe reductions in 'dry' beach widths and potential damages of assets located at the back of the beach. About 27 and 51 % of all Aegean beaches will retreat by more than their maximum width and 82 and 87 % will be completely overwhelmed by temporary flooding under short-term SLRs of 0.2 and 0.4 m, respectively

(Table 4).

In the case of a 0.4 m short-term SLR, up to 50 % and 86 % of all beaches fronting assets will be affected by beach retreat/erosion and temporary flooding, respectively (Fig. 6), whereas impacts will be more severe under higher levels (+0.6





m) (Table 4). The picture does not change much, even when projections are adjusted for the maximum overestimation observed in the results of the numerical models (by 19 %) when linear profiles were used (Section 4.2.1); for example, for a 0.4 m level rise, up to 37 and 84 % of all beaches fronting assets will be fully eroded and flooded, respectively.

The worst impacts are projected from the combined mean and short-term SLRs. In 2040, in the case that storm sea levels of 0.4 m are combined with a projected MSLR of 0.15 m (combined SLR of 0.55 m), 8 - 68 % of beaches are projected to be (at least temporarily) eroded and 21 - 91 % of beaches flooded. In 2100, superimposition of storm levels on the projected MSLRs will have devastating effects. A combined sea level rise of 1.1 m (e.g. a storm surge/wave set up of 0.6 m superimposed on a MSLR of 0.5 m (RCP4.5)) will have very severe impacts, indeed (Figs 6 and 7): 31 − 88 % of all beaches will be completely (at least temporarily) eroded (29 - 87 % of all beaches fronting assets) under the low and high mean projections of the ensemble, respectively, with 48 - 96 % of all beaches occasionally overwhelmed by flooding (Table 4).

A combined SLR of 1.3 m (RCP8.5, a MSLR of 0.7 m combined with a storm-induced coastal sea level of +0.6 m) represents a 'doom' scenario for the beaches of the Aegean Archipelago. Based on the low projections of the combined ensembles, about 75 % of beaches will be shifted landward (and/or drowned) to a distance equal to 50 % of their maximum 'dry' width, whereas about 39 % of all beaches will be (at least temporarily) completely eroded; 37 % of beaches fronting existing assets will be fully eroded and 54 % fully flooded.



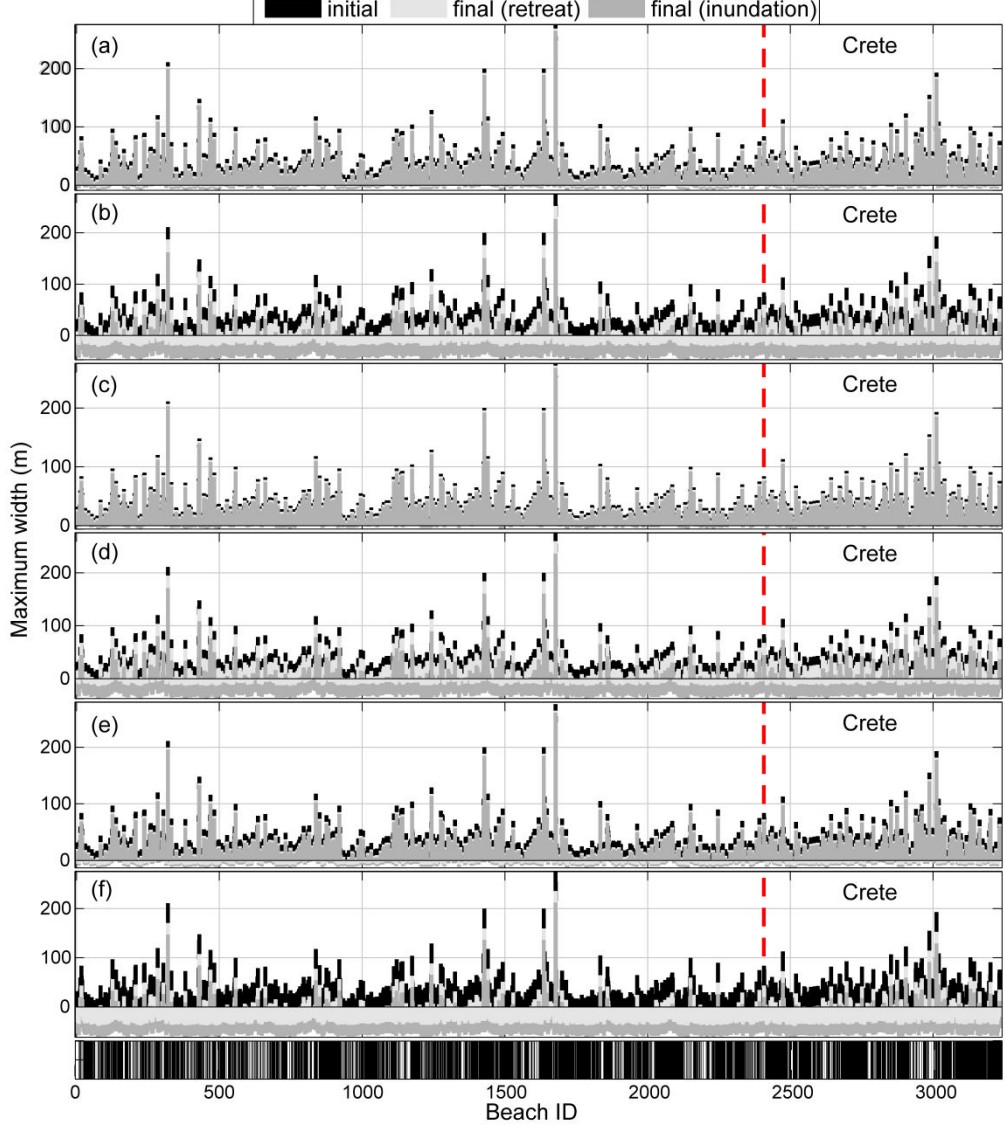

**Figure 6: Minimum and maximum retreat and flooding of Aegean Archipelago beaches for different SLR scenarios on the basis of the low and high means respectively of the ensemble projections. a) and b) Minimum and maximum retreat and flooding under a MSLR of 0.7 m. c) and d) Minimum and maximum retreat and flooding under a storm-induced level of +0.4 m. e) and f) Minimum and maximum retreat and flooding under a combined lon- and short-term SLR of 1.1 m (see text). Initial and final (after SLR) maximum beach widths are shown. Final widths < 0 show beaches lost or shifted landward or flooded by their entire maximum width. Black bars show currently existing infrastructure/assets fronted by beaches.**

10    According to the high ensemble estimates, it is projected that almost 91 % of all beaches will be completely eroded, at least for the time of the storm event (90 % of all beaches fronting assets); about 97 % of all beaches will also be flooded temporarily (Table 4).





**Figure 7: Projections of minimum beach retreat under a combined SLR of 1.1 m (see text), showing Aegean beaches projected to retreat by distances equal to different percentages of their initial maximum widths. In the inset, projections for the beaches of**
5 **Naxos (Cyclades) are shown in larger scale for detail.**





When projections are adjusted for overestimations observed in the validation exercise due to the use of linear profiles (Section 4.2.1), projected impacts are still devastating: 35 -89 % of all beaches will be (at least temporarily) completely

eroded and 48 – 86 % flooded, whereas according to the high estimates, 88 % and 96 % of all beaches fronting assets will be completely eroded and flooded, respectively.

It should be noted that projected impacts do not account for vertical land motions (Poulos et al., 2009). Moreover, since other significant erosion factors such as the diminishing coastal sediment supply are not considered, the above projections may actually underestimate future beach retreat/erosion and flooding at the Aegean Archipelago beaches.

**5 Discussion and Conclusions**

Analysis of the first systematic record of the geo-spatial characteristics of the beaches of the Aegean Archipelago has provided interesting findings. Aegean island beaches are narrow pocket beaches (59 % have maximum widths < 20 m), have limited terrestrial sediment supply and front a moderate (as a total) load of backshore development; about 81 % of the Aegean beaches currently front infrastructure/assets with 33 % fronting moderate to high coastal development. At the same

time, only 15 % of beaches are associated with some form of existing coastal protection. Moreover, at least 23 % of the Aegean beaches exhibit outcropping beachrocks at their beach faces. In addition to the safety risks that slippery beachrock surfaces might pose to beach users, beach face beachrocks can also degrade aesthetics, induce ecological changes and promote beach sediment erosion, decreasing the carrying capacity of the Archipelago beaches and constraining their potential as environments of leisure (Kontogianni et al., 2014).

Development and maintenance of the Aegean beaches may be (at least partially) controlled by hydrodynamics, as more and wider beaches occur along the southern and western coasts of the islands that are relatively protected from the prevailing N-NE waves. Maximum beach widths were found to correlate with sediment type and terrestrial sediment supply; sandy beaches are wider as are those associated with river outlets. With regard to the carrying capacity of the Archipelago beaches, these appear to have potential (as a total) for further development as environments of leisure; nevertheless, better spatial

distribution and a lengthening of the period of beach tourism should be considered.

The geo-spatial characteristics of the Aegean island beaches indicate that these may be considerably exposed to SLR. Our modeling results indeed project severe impacts under SLR from as early as 2040, particularly under the combined effects of the projected MSLR and storm-induced sea levels (Table 4). By 2100, impacts from combined MSLR and storm events could be devastating. For example, under a MSLR of 0.5 m (RCP4.5), a storm event inducing an additional 0.6 m rise is

projected to result in complete erosion of between 31 (minimum) and 88 % (maximum) of all beaches (29 - 87 % of beaches currently fronting coastal infrastructure and assets), at least temporarily. As a recent study projects substantial changes in the return periods of extreme sea levels for E. Mediterranean at the end of the 21st century (the current 1000 year event is



projected to occur once every 5 years, Vousdoukas et al. (submitted)), projected impacts on both beaches and coastal infrastructure/assets are worrying.

In the present study, SLR impacts have been estimated on the basis of decreases in 'dry' beach width, a critical parameter for the estimation of beach resilience and recreational use and value (e.g. Yang et al., 2012). Under increasing beach

erosion/retreat and flooding, the long-term recreational value of Aegean Archipelago beaches as well as the value of associated assets may fall considerably (e.g. Gopalakrishnan et al., 2011).

Against this background, it appears that plans to respond effectively to the projected beach erosion risk should be urgently drawn up with different adaptation options considered. Options based on the ecosystem approach should be the first to consider in order to protect both beaches and backshore ecosystems and infrastructure/assets (e.g. Peduzzi et al., 2013),

although 'hard' works might, in some cases, be deemed necessary. However, the significance of beaches as critical economic resources and the low effectiveness of 'hard' coastal works (e.g. breakwaters) to protect beaches from MSLR indicate that beach nourishment schemes will be required, at least for the most economically important beaches. As marine aggregates constitute the most suitable, but often scarce (Peduzzi, 2014) material for beach nourishment, particular care should be taken to ensure sustainability of marine aggregate deposits (Velegrakis et al., 2010). The significance of such deposits should be

certainly considered in future marine spatial plans as a matter of priority (see also EU Directive 2014/89/EU).

A primary tool to manage assets and economic activities at risk is the introduction of effective policies and regulation. An underpinning principle of coastal management under SLR should be the introduction of effective precautionary controls on future coastal development, e.g. through regulation that allocates buffer zones ('set-back' zones) behind retreating coastlines. It is submitted that, on the basis of our results, there is an urgent need to adopt/implement relevant regulation for the Aegean

Archipelago. A way forward could be the ratification by Greece of the 2009 ICZM Protocol to the Barcelona Convention, which prescribes set-back zones (Art. 8.2(a)) and has been already ratified by the EU (Council Decision 2010/631/EU). However, there are also challenges as e.g. those related to the set-back zone demarcation and risk allocation, (e.g. Sano et al., 2010; Gibbs et al., 2013) that can be only alleviated by publicly transparent criteria and decisions (e.g. Abbott, 2013). Assessments of beach retreat/erosion and flooding are required at various scales that are science-based and, at the same time,

accessible by coastal planners, managers, stakeholders and the wider public for the planning and smooth implementation of 'set-back' zones as well as the prioritization in the allocation of resources for adaptation.

The approach adopted in the present study to assess impacts of the SLR presents certain advantages. Existing methodologies/tools for rapid assessment of beach erosion and corresponding vulnerabilities to MSLR and extreme events at large scales (e.g. Hinkel et al., 2010; Ramieri et al., 2011; Khouakhi et al., 2013) have limitations stemming from (amongst

others): (a) their requirements for coastal Digital Elevation Models (DEMs) of high resolution/accuracy; and (b) the limited generally consideration for major controls (e.g. hydrodynamics). At the same time, advanced modeling approaches (e.g. Vousdoukas et al., 2016) in addition to detailed environmental information commonly require experienced operators and high computation costs that makes them impractical to coastal planners/managers (e.g. McLeod et al, 2010).



The present approach, which compares ranges of SLR induced beach retreat and flooding under different initial conditions and hydrodynamic forcing with beach maximum widths, is not limited by the resolution/accuracy of available coastal DEMs or the availability of detailed environmental information (e.g. Jiménez et al., 2012), and can be easily incorporated in other beach vulnerability tools (e.g. Alexandrakis et al., 2015) and used in areas with limited human resources. Nevertheless, there are also constraints. Projections are based on the assumption that beaches comprise inexhaustible sediment reservoirs, with no lateral sediment losses; cross-shore modeling obviously cannot resolve such issues. In addition, the approach is not designed to account for other erosion-controlling factors, such as: geological controls, coastal sedimentary budgets, and extreme event duration and sequencing (e.g. Gallop et al., 2012; Corbella and Stretch, 2012); the presence of artificial beach protection schemes and/or protecting nearshore ecosystems (e.g. Peduzzi et al., 2013); and the effects of coastal use (e.g. Bi et al., 2013). However, the aim of the exercise has not been to replace detailed modeling studies for individual beaches, but to provide ranges of beach erosion and flooding at a large (Archipelago) scale.

## Acknowledgements

This research has been co-financed by the European Union (European Social Fund – ESF) and Greek national funds through the Operational Program "Education and Lifelong Learning" of the National Strategic Reference Framework (NSRF) – Research Funding Program: THALES (Project ISLA). M.I. Vousdoukas acknowledges funding from the European Union Seventh Framework Programme FP7/2007–2013 under Grant Agreement No. 603864 (HELIX: 'High-End cLimate Impacts and eXtremes'; www.helixclimate.eu), as well as by the JRC institutional project Coastalrisk.

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
