# Peer review of "Assessment of island beach erosion due to sea level rise: The case of the Aegean Archipelago (Eastern Mediterranean)"

_Natural Hazards and Earth System Sciences, 2016_

## Referee Comment (RC1) · Anonymous Referee #1 · 9 Dec 2016

The manuscript details a study that (a) provides a record of the spatial characteristics of the Aegean Archipelago beaches (Greece); and (b) assesses the erosion and temporary inundation/flood risks of the Aegean beaches under different scenarios of SLR, using ensembles of cross-shore (1-D) morphodynamic models (validated, to an extent, by physical experiments) and empirically-derived estimations of wave run up-induced flooding. The manuscript is well written and referenced. It presents an interesting approach for the rapid assessment of the vulnerability of a large number of beaches at a regional level using minimal environmental information. In this sense, the presented approach is very useful and can breach the gap between coastal scientists/engineers and coastal managers and can be used in a variety of coastal regions. I recommend

publication following some clarification and minor editing corrections

I would like the authors to clarify in the manuscript the drivers/components of the short-term sea level increases. Do they refer to extreme sea levels due to storm surges and waves (e.g. is wave setup included) or to storm surge alone?

Pg.3, Lines 18 – 20 There are refs to US$ and then to €İt is confusing. Please clarify; this may be confusing for the reader.

Also Pg. 8, Lines 21 - 22 12 SLR scenarios are mentioned, but only 11 are detailed.

---

## Referee Comment (RC2) · Anonymous Referee #2 · 13 Dec 2016

I apologize for the lack of depth study that appears in this my revision. The manuscript deserves more attention, but the revision I had prepared was lost with the luggage during a flight, and I need to close this matter before leaving again. I hope chief editor and authors will forgive me! Pocket beaches' landscape and economic importance in small islands is huge and their extreme sensitivity to human interventions and climate change deserves greater attention. The subject matter of this work is of great interest but little developed in the literature. Impressive is the amount of analyzed data, and results overpass individual case and show what could be the effects of SLR on the entire economic sector of the Greek islands. We recommend the publication of this paper. Things that the authors could clarify: - The width of the beach has been extracted from Google Earth images and 4 operators obtained consistent results on 400 beaches, and considers irrelevant the influence of the tide (0,15 m) on its position. But it does not take into account the fact that the baric tides can have a much greater value and add up to the astronomical one. - Sediment texture can not be retrieved from satellite images for pixel size at ground. - No bathymetry data are presented for beaches, which affects the distance from the shore of the depth of closure, value that enters the erosion evaluation resulting in some SLR some models (e.g., Bruun). - Well-sorted sand was simulated, but data provided is D50, not sorting. - Beach rock exposure do not degrade beach aesthetics, its presence is considered a positive factor in Coastal Scenery Assessment.

---

## Author Comment (AC1) · 2 Jan 2017

We would like to thank Referee #1 for the useful suggestions. Please find below our response to his comments.

Comment 1: I would like the authors to clarify in the manuscript the drivers/components of the short-term sea level increases. Do they refer to extreme sea levels due to storm surges and waves (e.g. is wave setup included) or to storm surge alone?

Answer: This is a fine point by the reviewer. The drivers/components of the short-term sea level increases refer to extreme sea levels due to the combined effect of storm surges and wave set up. We agree with the reviewer that it needs to be clarified in the

text and we will make the necessary corrections in the revised manuscript.

Comment 2: Pg.3, Lines 18 – 20 There are refs to US\$ and then to € It is confusing. Please clarify; this may be confusing for the reader.

Answer: We agree with the reviewer and we will change the text to take into account this comment.

Comment 3: Also Pg. 8, Lines 21 - 22 12 SLR scenarios are mentioned, but only 11 are detailed.

Answer: A SLR scenario is missing by mistake from the context in the parentheses. We agree with the reviewer and we will correct the manuscript as suggested.
* * *

---

## Author Comment (AC2) · 2 Jan 2017

We would like to thank Referee #2 for the constructive comments. Please find below our response to his comments.

Comment 1: The width of the beach has been extracted from Google Earth images and 4 operators obtained consistent results on 400 beaches, and considers irrelevant the influence of the tide (0,15 m) on its position. But it does not take into account the fact that the baric tides can have a much greater value and add up to the astronomical one.

Answer: Beach width estimation is mainly affected from the shoreline position, which

is a dynamic coastal feature, showing continuous changes over time and space; being strongly connected with the beach morphodynamic processes. Recent research has shown that even in the case of "protected" beach systems (i.e. fronted by natural submerged reefs which act similar to artificial submerged breakwaters) for which data of high spatio-temporal resolution are available (i.e. 10 month period shoreline position data of hourly frequency), there is strong shoreline variability over time and space which in specific beach sections can reach up to 8 m (Velegrakis et al., 2016). This variability can be even higher in the case of satellite beach imagery of lower frequency (annual in the best case when it comes to Google earth application), for which different hydrodynamic conditions are evident (absence or presence of storminess, different tidal signal). Any method using satellite imagery, land based topographic methods or, even, video-imaging to provide positions of the shoreline has to deal with the fact that these positions may not delineate accurately the 'mean shoreline' e.g. the mean annual shoreline; this has been stated in the text (see page 7 lines 33-36). Accurate estimation of mean shoreline positions requires long time series of beach morphology of high temporal resolution, from which estimations of the mean shoreline position can be obtained (e.g. Aubrey, 1979). However, such information is rarely available, particularly at basin/Archipelago scale. Nevertheless, digitization through satellite imagery seems to be the most efficient way in identifying/estimating specific coastal features in order to provide a first assessment of the exposure to sea level rise over larger spatial scales, like the case of the Aegean archipelago beaches. If more accurate information becomes available in the future, this can be incorporated in the database, which is planned to become fully dynamic.

Velegrakis, A.F., Trygonis, V., Chatzipavlis, A.E., Karambas, Th., Vousdoukas, M.I., Ghionis, G., Monioudi I.N., Hasiotis, Th., Andreadis, O., Psarros, F., 2016. Shoreline variability of an urban beach fronted by a beachrock reef from video imagery. Natural Hazards DOI: 10.1007/s11069-016-2415-9.

Comment 2: Sediment texture cannot be retrieved from satellite images for pixel size

at ground.

Answer: The sediment texture (e.g. sand or gravel) was not retrieved from satellite images, it was assessed on the basis of the available photos on the Google Earth application and other available information collated from scientific literature/reports. To address this comment we will modify the text in order to make it clearer to the readership.

Comment 3: No bathymetry data are presented for beaches, which affects the distance from the shore of the depth of closure, value that enters the erosion evaluation resulting in some SLR some models (e.g., Bruun).

Answer: Given the large (Archipelago) scale of the application, the input data of the models for the evaluation of beach retreat, could not be based on in situ measurements. So we used linear profiles of a wide range of beach slopes. The distance from the shore of the depth of closure and the surf zone width, values necessary for the use of the analytical models, were estimated on the basis of the beach slope. The lack of accurate bathymetry data may enter some uncertainties in this point. However the validation of the models showed that the results of the models set with the equivalent linear profile were reasonably close to those of the physical experiments, and that the use of the models in an ensemble mode gave improved projections, with differences between models and experiments ranging from about 3 % to 11 % (see section 4.2.1). The aim of the exercise has not been to replace detailed modeling studies for individual beaches, but to provide ranges of beach erosion and flooding at a large (Archipelago) scale using minimum environmental information.

Comment 4: Well-sorted sand was simulated, but data provided is D50, not sorting.

Answer: One of the input data needed for the models is the median sediment size D50 (not sorting), this is the reason that D50 data are provided and not sorting. We will modify the text in order to make it clearer to the readership.

Comment 5: Beach rock exposure do not degrade beach aesthetics, its presence is considered a positive factor in Coastal Scenery Assessment.

Answer: According to literature beach rock can affect the actual size of a beach and diminish its carrying capacity (Vousdoukas et al, 2009). Moreover, the presence of weathered/deformed beachrock outcrops at the beachface, commonly colonised by assemblages of epilithic and borrowing organisms (Brattström, 1992) that form a 'slippery' mat, can make the access to the sea difficult, or even dangerous, and degrade the aesthetics and amenity value of the beach and, thus, affect its touristic potential (Vousdoukas et al, 2009). Beachrock formation/outcropping may also result in increased biodiversity in the coastal zone, since beachrock outcrops can create habitats suitable for colonisation by hard-substrate species (e.g. corals, molluscs, algae and annelid worms) (Brattström, 1992; Vousdoukas, Velegrakis, & Plomaritis, 2007). However, it is guestionable whether the overall effect on the coastal ecology would be beneficial, particularly in view of the biodiversity losses in soft-substrate species (Brown, 1982). Beach aesthetics/scenery may suffer by the presence of beachrocks, as beachrockinfected beaches do not comply with the widely-recognisable beach model (long and wide beaches consisting of light coloured sands). With regard to the tourists' perceptions on this subject, a contingent valuation study among European tourists showed that although the majority of tourists were not previously aware of beachrock phenomenon, half of them paid notice to the hard coastal sedimentary formations. Survey respondents believe that the authorities should undertake precautionary measures and that European Union should increase research funding in order to avoid further beachrock expansion. Actually, almost half of the respondents would be willing to pay an annual tax in the range of 13.2-16.4 € per household in order to contribute to this effort (Kontogianni et al., 2014). We may add some refs in order to address the recommendation, but due to space limitations we do not think that further documentation of this subject would be beneficial to the manuscript.

Brattström, H., 1992. Marine biological investigations in the Bahamas. 22. Littoral

zonation at three Bahamian beachrock localities. Sarsia 77, 81-109.

Brown, B., 1982. Spatial and temporal distribution of a deposit-feeding polychaete on a heterogeneous tidal flat. Journal of Experimental Marine Biology and Ecology 65(3), 213-227.

Kontogianni, A., Damigos, D., Tourkolias, C., Vousdoukas, M., Velegrakis, A., Zanou, B., & Skourtos, M., 2014. Eliciting beach users' willingness to pay for protecting European beaches from beachrock processes. Ocean & Coastal Management 98, 167-175.

Vousdoukas, M.I., Velegrakis, A.F., & Plomaritis, T.A., 2007. Beachrock occurrence, characteristics, formation mechanisms and impacts. Earth-Science Reviews 85, 23-46.

Vousdoukas, M.I., Velegrakis, A.F., Kontogianni, A., Makrykosta, E.N., 2009. Implications of the cementation of beach sediments for the recreational use of the beach. Tour. Manag. 30 (4), 544-552.

---

## Author Comment (AC3) · 2 Jan 2017

We would like to thank the referees for the useful suggestions and constructive comments. Please find below our response to their comments

**Referee #1**

Comment 1: I would like the authors to clarify in the manuscript the drivers/components of the short-term sea level increases. Do they refer to extreme sea levels due to storm surges and waves (e.g. is wave setup included) or to storm surge alone?

Answer: This is a fine point by the reviewer. The drivers/components of the short-term sea level increases refer to extreme sea levels due to the combined effect of storm

surges and wave set up. We agree with the reviewer that it needs to be better clarified in the text and we will do so in the revised manuscript.

Comment 2: Pg.3, Lines 18 - 20 There are refs to US\$ and then to  $\in$  It is confusing. Please clarify; this may be confusing for the reader.

Answer: We will make the necessary changes to address this comment.

Comment 3: Also Pg. 8, Lines 21 - 22 12 SLR scenarios are mentioned, but only 11 are detailed.

Answer: This SLR scenario was missing by mistake. We will make the necessary corrections.

**Referee #2**

Comment 1: The width of the beach has been extracted from Google Earth images and 4 operators obtained consistent results on 400 beaches, and considers irrelevant the influence of the tide (0,15 m) on its position. But it does not take into account the fact that the baric tides can have a much greater value and add up to the astronomical one.

Answer: We certainly agree with the referee that beach width estimations from satellite snapshots may not represent mean conditions, as we have stated in the text (Section 3.1, beginning of page 7). Beach width estimations are controlled by the shoreline positions, which are dynamic coastal feature, showing large spatial and temporal variability; and being strongly controlled by the beach morphodynamic processes. Recent detailed research on one of the Aegean Archipelago beaches (Ammoudara, N. Crete) for which data of high spatio-temporal resolution are available (i.e. 10 month period hourly shoreline positions),has shown high shoreline position variability (up to 6 m, or about 10-12 % of the maximum width (Velegrakis et al., 2016). We believe that any method to record beach widths using satellite imagery snapshots, one-off land based topographic and/or LIDAR surveys or, even, video-imaging of limited duration may not

provide synoptic information on the 'mean' beach conditions. Accurate estimation of mean shoreline positions requires long time series of high temporal resolution, from which estimations of the mean shoreline position could be obtained. However, such information is rarely available, particularly at the basin/Archipelago scale. Therefore, satellite imagery information appears to be the only alternative, which we deem adequate for a first assessment of the beach exposure to sea level rise over larger spatial scales. We will rewrite the text to clarify further the above issue.

Comment 2: Sediment texture cannot be retrieved from satellite images for pixel size at ground.

Answer: The sediment texture (e.g. sand or gravel) was not retrieved from satellite images, it was assessed on the basis of the available photos on the Google Earth application and other available information collated from scientific literature/reports. To address this comment there will be further clarification in the text to make it clearer to the readership.

Comment 3: No bathymetry data are presented for beaches, which affects the distance from the shore of the depth of closure, value that enters the erosion evaluation resulting in some SLR some models (e.g., Bruun).

Answer: Given the large (Archipelago) scale of the application, the input data of the models for the evaluation of beach retreat, could not be based on in situ measurements. So we used linear profiles of a wide range of beach slopes. The distance from the shore of the depth of closure and the surf zone width, values necessary for the use of the analytical models, were estimated on the basis of the beach slope. The lack of accurate bathymetry data may introduce some uncertainty. However the validation of the models showed that the results of the models set with the equivalent linear profile were reasonably close to those of the physical experiments, and that the use of the models in an ensemble mode gave improved projections, with differences between models and experiments ranging from about 3 % to 11 % (see section 4.2.1). The

aim of the exercise has not been to replace detailed modeling studies for individual beaches, but to provide ranges of beach erosion and flooding at a large (Archipelago) scale using minimum environmental information. We will make the necessary clarifications in the revised manuscript

Comment 4: Well-sorted sand was simulated, but data provided is D50, not sorting.

Answer: One of the input data needed for the models is the median sediment size D50 (not sorting), this is the reason that D50 data are provided and not sorting. We will modify the text in order to make it clearer to the readership.

Comment 5: Beach rock exposure do not degrade beach aesthetics, its presence is considered a positive factor in Coastal Scenery Assessment.

Answer: In this point we do not agree with the referee. Beachrocks not only can affect the actual widths of a sandy beach as they can promote beach sediment erosion and outcropping of the initially buried beachrocks (see Vousdoukas et al., 2007, Vousdoukas et al, 2009a), but also affect perceptions regarding the beach. The presence of weathered/deformed beachrock outcrops at the beachface, commonly colonised by assemblages of epilithic and borrowing organisms (Brattström, 1992) that form a 'slippery' mat, can make the access to the sea difficult, or even dangerous, and degrade the aesthetics and amenity value of the beach and, thus, affect its touristic potential (Vousdoukas et al. 2009b). Beachrock formation/outcropping may also change the biodiversity, since beachrock outcrops create habitats suitable for colonisation by hardsubstrate species (e.g. corals, molluscs, algae and annelid worms) (Brattström, 1992; Vousdoukas et al., 2007; 2012). However, it is questionable whether the overall effect on the coastal ecology would be beneficial, particularly in view of the biodiversity losses in soft-substrate species (Brown, 1982). Beach aesthetics/scenery may also suffer by the presence of beachrocks, as beachrock beaches do not comply with the widely-recognisable beach model (long and wide beaches consisting of light coloured sands).

With regard to the tourists' perceptions on this subject, a contingent valuation study among European tourists showed that although the majority of tourists were not previously aware of beachrock phenomenon, half of them paid notice to the hard coastal sedimentary formations. Survey respondents believe that the authorities should undertake precautionary measures and that European Union should increase research funding in order to avoid further beachrock expansion. Actually, almost half of the respondents would be willing to pay an annual tax in the range of 13.2-16.4  $\in$  per household in order to contribute to this effort (Kontogianni et al., 2014).

We will clarify further the issue and add some refs in order to address the comment but due to space limitations we do not think that detailed documentation would be beneficial to the manuscript.

**References**

Brattström, H., 1992. Marine biological investigations in the Bahamas. 22. Littoral zonation at three Bahamian beachrock localities. Sarsia 77, 81-109.

Brown, B., 1982. Spatial and temporal distribution of a deposit-feeding polychaete on a heterogeneous tidal flat. Journal of Experimental Marine Biology and Ecology 65(3), 213-227.

Kontogianni, A., Damigos, D., Tourkolias, C., Vousdoukas, M., Velegrakis, A., Zanou, B., & Skourtos, M., 2014. Eliciting beach users' willingness to pay for protecting European beaches from beachrock processes. Ocean & Coastal Management 98, 167-175.

Velegrakis, A.F., Trygonis, V., Chatzipavlis, A.E., Karambas, Th., Vousdoukas, M.I., Ghionis, G., Monioudi I.N., Hasiotis, Th., Andreadis, O., Psarros, F., 2016. Shoreline variability of an urban beach fronted by a beachrock reef from video imagery. Natural Hazards DOI: 10.1007/s11069-016-2415-9.

Vousdoukas, M.I., Velegrakis, A.F., & Plomaritis, T.A., 2007. Beachrock occurrence, characteristics, formation mechanisms and impacts. Earth-Science Reviews 85, 23-

**46.**

Vousdoukas, M., Velegrakis, A.F. and Karambas, Th., 2009. Morphology and sedimentology of a beachrock-infected beach: Vatera Beach, Lesbos, Greece. Continental Shelf Research 29, 1937–1947.

Vousdoukas, M.I., Velegrakis, A.F., Kontogianni, A., Makrykosta, E.N., 2009. Implications of the cementation of beach sediments for the recreational use of the beach. Tourism Managemant, 30 (4), 544-552.

Vousdoukas, M.I., A.F. Velegrakis, M. Paul, C. Dimitriadis, E. Makrykosta, D. Koutsoubas, 2012. Field observations and modeling of wave attenuation over colonized beachrocks. Continental Shelf Research 48, 100-109.

---

## Author Response (AR1)

**Final response to the Referees' comments**

First, we would like to thank the editor and the referees for the useful suggestions and constructive comments. To respond to these comments, we have made several modifications to the manuscript and we added some new refs. Please find below our response to their comments and a marked-up manuscript version showing the changes made.

Editor

*Comment 1: Figure 1 has no legend for the terrestrial topography*

Answer: This is a fine point by the editor. We have changed Figure 1 as suggested (section 2.2, page 5).

Referee #1

*Comment 1: I would like the authors to clarify in the manuscript the drivers/components of the short-term sea level increases. Do they refer to extreme sea levels due to storm surges and waves (e.g. is wave setup included) or to storm surge alone?*

Answer: This is a fine point by the reviewer. The drivers/components of the short-term sea level increases refer to extreme sea levels due to the combined effect of storm surges and wave set up. We agree with the reviewer that it needs to be better clarified in the text and we have made the necessary changes in the revised manuscript (section 2.3, page 6, lines 12-13, section 3.2, page 9, lines 10-11 and 15-17, section 4.2.2, page 19, lines 8-9)

*Comment 2: Pg.3, Lines 18 – 20 There are refs to US$ and then to € It is confusing. Please clarify; this may be confusing for the reader.*

Answer: Changed as suggested (section 2.1, page 3, line 22)

*Comment 3: Also Pg. 8, Lines 21 - 22 12 SLR scenarios are mentioned, but only 11 are detailed.*

Answer: The examined SLR scenarios are eleven. We made the necessary correction in the revised manuscript (section 3.2, page 9, line 8)

Referee #2

*Comment 1: The width of the beach has been extracted from Google Earth images and 4 operators obtained consistent results on 400 beaches, and considers irrelevant the influence of the tide (0,15 m) on its position. But it does not take into account the fact that the baric tides can have a much greater value and add up to the astronomical one.*

Answer: We certainly agree with the referee that beach width estimations from satellite snapshots may not represent mean conditions, as we have stated in the text (Section 3.1, beginning of page 7). Beach width estimations are controlled by the shoreline positions, which are dynamic coastal feature, showing large spatial and temporal variability; and being strongly controlled by the beach morphodynamic processes. Recent detailed research on one of the Aegean Archipelago beaches (Ammoudara, N. Crete) for which data of high spatio-temporal resolution are available (i.e. 10 month period hourly shoreline positions), has shown high shoreline position variability (up to 6 m, or about 10-12 % of the maximum width (Velegrakis et al., 2016). We believe that any method to record beach widths using satellite imagery snapshots, one-off land based topographic and/or LIDAR surveys or, even, video-imaging of limited duration may not provide synoptic information on the 'mean' beach conditions. Accurate estimation of mean shoreline positions requires long time series of high temporal

resolution, from which estimations of the mean shoreline position could be obtained. However, such information is rarely available, particularly at the basin/Archipelago scale. Therefore, satellite imagery information appears to be the only alternative, which we deem adequate for a first assessment of the beach exposure to sea level rise over larger spatial scales. We have modified the text in order to clarify further the above issue (section 3.1, page 7, lines 3-11).

*Comment 2: Sediment texture cannot be retrieved from satellite images for pixel size at ground.*

Answer: The sediment texture (e.g. sand or gravel) was not retrieved from satellite images, it was assessed on the basis of the available photos on the Google Earth application and other available information collated from scientific literature/reports. To address this comment we modified the text to make it clearer to the readership (section 3.1, page 7, lines 24-27).

*Comment 3: No bathymetry data are presented for beaches, which affects the distance from the shore of the depth of closure, value that enters the erosion evaluation resulting in some SLR some models (e.g., Bruun).*

Answer: Given the large (Archipelago) scale of the application, the input data of the models for the evaluation of beach retreat, could not be based on in situ measurements. So we used linear profiles of a wide range of beach slopes. The distance from the shore of the depth of closure and the surf zone width, values necessary for the use of the analytical models, were estimated on the basis of the beach slope. The lack of accurate bathymetry data may introduce some uncertainty. However the validation of the models showed that the results of the models set with the equivalent linear profile were reasonably close to those of the physical experiments, and that the use of the models in an ensemble mode gave improved projections, with differences between models and experiments ranging from about 3 % to 11 % (see section 4.2.1). The aim of the exercise has not been to replace detailed modeling studies for individual beaches, but to provide ranges of beach erosion and flooding at a large (Archipelago) scale using minimum environmental information.
The necessary clarification is made in the revised manuscript (section 3.2, page 8, lines 10-13).

*Comment 4: Well-sorted sand was simulated, but data provided is D50, not sorting.*

Answer: One of the input data needed for the models is the median sediment size D50 (not sorting), this is the reason that D50 data are provided and not sorting. We modified the text in order to make it clearer to the readership (section 4.2.1, page 13, line 24).

*Comment 5: Beach rock exposure do not degrade beach aesthetics, its presence is considered a positive factor in Coastal Scenery Assessment.*

Answer: In this point we do not agree with the referee. Beachrocks not only can affect the actual widths of a sandy beach as they can promote beach sediment erosion and outcropping of the initially buried beachrocks (see Vousdoukas et al., 2007, Vousdoukas et al, 2009a), but also affect perceptions regarding the beach. The presence of weathered/deformed beachrock outcrops at the beachface, commonly colonised by assemblages of epilithic and borrowing organisms (Brattström, 1992) that form a 'slippery' mat, can make the access to the sea difficult, or even dangerous, and degrade the aesthetics and amenity value of the beach and, thus, affect its touristic potential (Vousdoukas et al, 2009b). Beachrock formation/outcropping may also change the biodiversity, since beachrock outcrops create habitats suitable for colonisation by hard substrate species (e.g. corals, molluscs, algae and annelid worms) (Brattström, 1992; Vousdoukas et al., 2007; 2012). However, it is questionable whether the overall effect on the coastal ecology would be beneficial, particularly in view of the biodiversity losses in soft-substrate species (Brown, 1982). Beach aesthetics/scenery may also suffer by the presence of beachrocks, as beachrock beaches do not comply with the widely-recognisable beach model (long and wide beaches consisting of light coloured sands). With regard to the tourists' perceptions on this subject, a contingent valuation study among European tourists showed that although the majority of tourists were not previously aware of beachrock phenomenon, half of them paid notice to the hard coastal sedimentary formations. Survey respondents believe that the authorities should undertake precautionary measures and that European Union should increase research funding in order to avoid further beachrock expansion. Actually,

almost half of the respondents would be willing to pay an annual tax in the range of 13.2-16.4 €per household in order to contribute to this effort (Kontogianni et al., 2014).

We clarified further the issue by adding some refs in order to address the comment but due to space limitations we do not think that detailed documentation would be beneficial to the manuscript (section 5, page 22, lines 17-19).

[revised manuscript text omitted]

**Σχόλιο [I15]:** Refs added in order to address the Comment 5 of Referee #2.

**Σχόλιο [a16]:** Added that reference to address further the Comment 1 of Referee #1.